# A Comparison between Core Stability Exercises and Muscle Thickness Using Two Different Activation Maneuvers

**DOI:** 10.3390/jfmk9020070

**Published:** 2024-04-11

**Authors:** Ioannis Tsartsapakis, Ioanna Bagioka, Flora Fountoukidou, Eleftherios Kellis

**Affiliations:** Laboratory of Neuromechanics, Department of Physical Education and Sport Sciences at Serres, Aristotle University of Thessaloniki, 62100 Serres, Greece

**Keywords:** core stability, bracing, hollowing, ultrasound, muscle thickness

## Abstract

Core stability training is crucial for competitive athletes, individuals who want to improve their health and physical performance, and those undergoing clinical rehabilitation. This study compared the ultrasound (US) muscle thickness of the abdominals and lumbar multifidus (LM) muscles between seven popular trunk stability exercises performed using hollowing and bracing maneuvers. Forty-four healthy young adults, aged between 21 and 32 years, performed a plank, bird dog, beast crawl, dead bug, Pilates tap, bridge, and side planks using the bracing and the hollowing maneuver. The thickness of the transversus abdominis (TrA), internal oblique (IO), and LM muscles was measured simultaneously using two ultrasound machines. Analysis of variance designs indicated that during hollowing, the bird dog and side plank exercises resulted in the greatest increase in the muscle’s relative thickness overall. The relative thickness of all muscles was significantly greater (*p* < 0.001) during hollowing (22.7 ± 7.80 to 106 ± 24.5% of rest) compared to bracing (18.7 ± 7.40 to 87.1 ± 20.9% of rest). The TrA showed the greatest increase in thickness (*p* < 0.001) compared to the IO and LM. Additionally, the IO had a greater increase in thickness (*p* < 0.001) than the LM. In conclusion, our findings indicate that the bird dog and side plank exercises, when performed with hollowing, showed the most significant total muscle thickness increase. Notably, the hollowing maneuver enhances the thickness of the TrA, IO, and LM muscles more than the bracing maneuver. This contributes to the discussion on optimal strategies for dynamic core stabilization.

## 1. Introduction

Many scientists acknowledge the importance of core stability training in stabilizing and building strength for both sports and daily activities [1]. A meta-analysis by Dong et al. [2] concluded that core stability training had a large effect on athletes’ core strength and balance but little effect on specific athletic performance. In addition, core stability training was more effective than general exercise at reducing pain and improving back-specific functional status in patients with low back pain [3]. Core stability training frequently refers to exercises which aim to activate the muscles of the trunk having a stabilizing function [4]. Such exercises are utilized for the enhancement of sport performance [1,5], improvements in health and physical performance [6,7], and rehabilitation [8,9,10]. Hence, choosing the right exercises is an important step for an effective core stability training program and is an issue that often arises in this context.

Studies have utilized ultrasound (US) imaging to measure the thickness of the deeper abdominal and spinal muscles at rest and during exercise, which is an indirect index of muscle contraction/activation during exercise [11,12,13]. The studies above commonly conclude that an ultrasound measurement of the trunk muscle thickness is a reliable method that can provide useful information about the function of these muscles.

Research studies have compared various core stability exercises and their variations, such as the plank, supine position, side plank, dead bug, bridge, and bird dog, to determine which one activates the core muscles the most [14,15,16,17,18,19]. However, each study suggested that a different exercise (e.g., plank, side plank, dead bug, bridge, bird dog) can achieve a greater increase in muscle thickness than the others. For example, Imai et al. [19] found that the plank exercise activates all the muscles of the trunk to a greater extent. Emami et al. [17] found that the bird dog exercise is most effective for activating deep spinal muscles, while Moghadam et al. [14] argue that the bridge exercise leads to the greatest increase in trunk muscle thickness. Consequently, data on core training programs and their efficacy in everyday, mild and recreational exercise are incomplete. Finally, it appears that each muscle is activated to varying degrees depending on the core exercise performed [16,19]. Therefore, the literature does not propose a specific order of exercises for the total activation of all core muscles.

The performance of trunk stabilization exercises is frequently combined with additional maneuvers, namely, abdominal hollowing or bracing. The former focuses on the contraction of specific muscles, such as the transversus abdominis (TrA), internal oblique (IO), and lumbar multifidus (LM), which pull the navel toward the lumbar spine. Conversely, bracing exercises require one to push their abdomen outwards and engage all the core muscles (rectus abdominis—RA; external oblique—EO; iliocostalis lumborum—IL) simultaneously [20,21]. However, there are inconsistencies in the evidence regarding the differences in the activation of the deep spinal muscles between these two types of maneuvers. Previous studies using ultrasound in supine [22,23], sitting [23], and standing [24] postures reported an increase in TrA thickness by almost two-fold during hollowing in comparison to bracing. Furthermore, research has shown that IO and EO muscle thickness was greater during hollowing as opposed to bracing [23]. A study [14] found conflicting findings. Moghadam et al. [14] concluded that the greatest increase in TrA thickness occurred when bracing as opposed to hollowing. Moreover, studies conducted using electromyography (EMG) have produced contradictory results. Some studies have shown that hollowing results in a greater activation of the core muscles [25,26], while others have found the opposite [27]. Based on the above studies, it is unclear which of the two maneuvers is better than the other in terms of activating the trunk stabilizers. This means that exercise professionals cannot currently decide which of these maneuvers can activate certain muscles of the spine.

Over the past decade, numerous scientists have made a concerted effort to identify exercises that maximize the activation of deep core muscles [14,15,28,29]. The results of these studies vary, possibly due to the lack of a specific exercise protocol for core stability and differing goals among researchers from various scientific fields. Hence, the aim of this study was to compare the US thickness of the TrA, IO, and LM between seven popular core stability exercises, using the bracing or hollowing maneuvers. It was hypothesized that the order of exercises in terms of the relative muscle thickness increase will be different for the TrA, IO, and LM. Additionally, we hypothesize that one of the seven protocol exercises, when combined with the hollowing maneuver, will result in a significant difference in overall relative muscle thickness compared to the other exercises. The abdominal hollowing maneuver will produce a significantly greater thickness separately on each of the TrA, IO, and LM muscles in all exercises compared to the abdominal bracing maneuver.

## 2. Materials and Methods

The GPower analysis [Ver. 3.1.9.7; Heinrich-Heine-Universität Düsseldorf, Düsseldorf, Germany; http://www.gpower.hhu.de/ (accessed on 14 February 2024)] revealed that to achieve a power of 0.98, a total sample size of 30 was necessary, given an effect size of 0.35 and an alpha level of 0.05.

### 2.1. Participants

Participants were recruited through advertisements on internet exercise sites and volunteered to participate. Verbal interviews were conducted with all subjects, and those who had been engaging in recreational exercise for at least two years were included in the study. The study excluded individuals who were competitive-level professional or amateur athletes, regular users of the Pilates method of exercise, and those who were part of core training programs. Additionally, volunteers were excluded if they had recent or chronic symptoms of low back pain, shoulder, and knee injuries. Interviews were conducted with all participants during the day between September and November 2023. Those who did not meet the inclusion criteria were excluded. This study selected a sample of 44 healthy young adults (18 men), aged 27.0 ± 3.03 years (ranging from 21 to 32 years old). Participants were advised to avoid strenuous exercise for at least three days prior to the test and to abstain from alcohol for at least two days before the test. The participants provided their written consent, and the protocol was approved by Ethics University Committee (ERC-007/2024, 12 March 2024.). The characteristics of the participants are presented in Table 1.

### 2.2. Instruments

Two computerized ultrasound (US) systems (Aloka ProSound SSD 3500 SV, Aloka Co. Ltd., Tokyo, Japan) were employed to measure the thickness of the deep spinal muscles TrA, IO, and LM simultaneously (Figure 1). The transducer head, with a length of 6 cm, operated at a frequency of 13 MHz. Ultrasound gel was applied to enhance the contact area and reduce the need for inward probe pressure during muscle thickness measurement [30].

### 2.3. Ultrasound Muscles Thickness Measurements

The first probe was used to measure the thickness of the TrA and IO muscles, and another probe was used to measure the thickness of the LM. For the TrA and IO, the probe was positioned 2.5 cm above the iliac crest and along the axillary line [13]. The probe was oriented in the transverse plane, perpendicular to the muscle fibers, and situated 2 cm to the left of the center of the US image when the subject was relaxed. The TrA is the deepest layer of the abdominal musculature, followed by the IO and then the EO. The thickness of the TrA and IO muscles was determined by measuring the distance between their superior and inferior fascia while the subjects were either bracing or hollowing. For the LM, the probe was positioned perpendicular to the spine on the lower back, specifically at the L2–L4, L4–L5 [31] intervertebral spaces. The probe was oriented in the longitudinal plane, parallel to the muscle fibers. The thickness of the LM was measured as the distance between the superior and inferior fascia of the muscle. The relative thickness during each exercise was calculated as a percentage of the resting thickness for each muscle and exercise using the formula [32].
$$\frac{\mathrm{contraction}\;-\;\mathrm{rest}\;}{\;\mathrm{rest}\;}\times 100$$

### 2.4. Exercise Protocol

Prior to the main measurement, standard preparation was administered to all participants. Significant effort was dedicated to ensuring that all participants accurately comprehended both maneuvers through the use of real-time ultrasound imaging (RUSI). Once participants were able to perform both maneuvers accurately without the use of RUSI, we proceeded with the main measurement. The thickness of the right relaxed abdominals (TrA/IO) and LM was measured in the supine and prone positions, respectively. The selected exercises for comparison were based on previous studies [14,15,16,17,18,19] that demonstrated their effectiveness in activating the deep muscles of the trunk. Then, participants performed seven exercises using the bracing and hollowing technique, in random order as follows (Figure 2):(a)The plank exercise with elbow support and one arm extended: this variation of the traditional plank exercise primarily targets the core muscles, including the rectus abdominis, internal and external obliques, erector spinae, transverse abdominis, and lumbar multifidus.(b)The bird dog exercise: this bodyweight exercise targets the muscles of the back, abdominals, hips, and glutes.(c)The shoulder bridge exercise with one leg extended: this variation of the traditional shoulder bridge exercise targets the lower back, abs, glutes, and hamstrings.(d)The Pilates toe tap exercise: this well-known Pilates exercise targets the transverse abdominis, rectus abdominis, and obliques, working to strengthen the core.(e)The dead bug exercise: this bodyweight exercise targets several muscles, including the rectus abdominis, internal obliques, external obliques, transverse abdominis, multifidus, erector spinae, and the pelvic floor.(f)The beast crawling exercise: this exercise is a variation of the static Beast Hold exercise. It targets the core muscles, including the transverse abdominis, rectus abdominis, internal and external obliques, erector spinae, and spinal back muscles, as well as the shoulders and quads.(g)The side plank exercise with extended arm: this variation of the traditional side plank exercise targets the oblique abdominal and lateral hip muscles.

Participants initially assumed the basic position for each exercise. Upon the examiner’s command, they inhaled and executed each exercise. A metronome controlled the rhythm of the movement of the body parts. The thickness of the three muscles was measured during exhalation and the voluntary movement of the body part. Three measurements were taken for each method, and the average was recorded.

For detailed instructions, please refer to the Appendix A.

### 2.5. Statistical Analysis

Statistical analyses were performed using IBM SPSS Statistics ver. 29.0 (IBM Co., Armonk, NY, USA). The Kolmogorov–Smirnov test was utilized to confirm the normal distribution of the collected data. To analyze the difference in muscle thickness, a three-way (3 muscles × 7 exercises × 2 maneuvers) repeated measures analysis of variance (ANOVA) test was applied. Post hoc Bonferroni tests were conducted to examine the pairwise differences among the muscles, maneuver methods, and exercises. In addition, partial eta squared (η*_p_*^2^) values were computed to indicate the effect size. The Greenhouse–Geisser correction was applied when the assumption of sphericity was violated. The level of significance was set at *p* < 0.05.

## 3. Results

### Ultrasound Relative Thickness

Table 2 presents the mean descriptive scores (±SD) for relative muscle thickness in the US for each muscle, exercise, and maneuver. The three-way (muscles, exercises, maneuvers) interaction was not statistically significant (*F*_(12.516)_ = 1.566, *p* = 0.098, η*_p_*^2^ = 0.035).

The muscle and exercise interaction effect was statistically significant (*F*_(12.256)_ = 4.242, *p* < 0.001, η*_p_*^2^ = 0.090). Post hoc tests showed the TrA’s relative thickness was greater compared to the IO and LM in all seven exercises (*p* < 0.001, *p* < 0.001). Additionally, the relative thickness of the IO was significantly greater than that of the LM in all seven exercises (*p* < 0.001) (Table 2).

Furthermore, the Bonferroni post hoc analysis revealed that the TrA had a significantly greater relative thickness during the bird dog exercise compared to the plank exercise (*p* = 0.006) and the bridge exercise (*p* = 0.027). In the ranking of exercises based on the relative thickness of the TrA, the bird dog exercise ranks first, followed by the side plank, beast crawl, dead bug, and toe tap exercises. The bridge and plank exercises are ranked last (Figure 3).

Regarding the IO muscle, the Bonferroni post hoc analysis showed that the side plank exercise had a statistically significant greater relative thickness compared to the dead bug (*p* < 0.001), bridge (*p* < 0.001), plank (*p* = 0.006), and toe tap (*p* = 0.007) exercises. The statistical analysis revealed that the bird dog exercise differed significantly in relative IO thickness from the dead bug (*p* < 0.001), toe tap (*p* = 0.011), plank (*p* = 0.012), and bridge (*p* = 0.026) exercises. Additionally, the relative thickness of the IO in the beast crawl exercise differed significantly from that in the toe tap exercise (*p* = 0.045) (see Figure 3). According to the ranking of exercises based on the relative thickness of the IO, the side plank exercise is ranked first, followed by the bird dog exercise in second place, and the beast crawl exercise in third place. The plank, bridge, toe tap, and dead bug exercises follow (Figure 3).

In relation to the LM muscle, the side plank exercise had a statistically significant greater relative thickness compared to the beast crawl (*p* < 0.001) and the plank (*p* = 0.027), according to the Bonferroni post hoc analysis. Additionally, the dead bug exercise had a statistically significant greater relative thickness compared to the beast crawl (*p* = 0.008) and the plank (*p* = 0.032) exercises (refer to Figure 3). The exercises were ranked based on the relative thickness of the LM muscle, with the side plank and dead bug exercises being ranked first, followed by the toe tap, bridge, and bird dog exercises. The plank and beast crawl exercises were ranked last (all the results of statistically significant relationships between the exercises are shown in Figure 3).

A statistically significant exercise by the maneuver’s interaction effect was found, indicating that all seven exercises had a significantly higher mean relative thickness during hollowing (*p* < 0.001) compared to bracing (*F*_(6.258)_ = 2.262, *p* = 0.038, η*_p_*^2^ = 0.050). The Bonferroni post hoc test showed that the bird dog exercise and the side plank exercise with the hollowing maneuver had a statistically significant greater relative thickness from the plank (*p* < 0.001, *p* < 0.001), bridge (*p* = 0.006, *p* = 0.005), and toe tap exercises (*p* < 0.001, *p* = 0.005), respectively. Moreover, both exercises exhibited no significant difference in relative thickness from each other, as well as from the dead bug and beast crawl exercises (*p* > 0.050) (Figure 4). The exercises were ranked according to their total mean relative thickness score. The bird dog and side plank exercises were ranked first, followed by the dead bug and beast crawl exercises. The plank, toe tap, and bridge exercises were ranked last, as shown in Figure 4.

A significant muscle by maneuver interaction was also found (*F*_(2.86)_ = 38.373, *p* < 0.001, η*_p_*^2^ = 0.472). The hollowing maneuver resulted in a statistically significant increase in muscle relative thickness compared to the bracing maneuver for all three muscles (*p* < 0.001, *p* < 0.001, *p* < 0.001) (Figure 5). The Bonferroni post hoc tests confirmed that the TrA’s relative thickness was greater than the IO’s (*p* < 0.001) and LM’s (*p* < 0.001) thickness, for each maneuver type. Moreover, the relative thickness of the IO was significantly greater than that of the LM (*p* < 0.001) (Figure 5)

## 4. Discussion

Core stability training is important for maintaining the health of individuals who exercise and preventing lumbar spine disorders [33,34]. For the first hypothesis of this study, the results showed that the order of exercises from the highest to lowest relative thickness ratio differed between muscles. In particular, the TrA relative thickness was greater during the bird dog exercise and the lowest during the plank exercise (Figure 3). According to several authors [35,36,37], the TrA acts as a lumbar corset and is the primary stabilizer of the lumbar spine and pelvis. The TrA is activated prior to each limb movement and is activated in anticipation of lower limb movement [35,38,39]. The bird dog exercise involves moving one foot and the opposite hand simultaneously [17]. This movement shifts the body’s weight to the other limbs, which can affect the strength and stability of the trunk muscles and the pressure on the spine. From a biomechanical point of view, the bird dog exercise engages the anterior deltoid muscles as an agonist for shoulder flexion, while the gluteus maximus muscles act as an agonist for hip extension. The posterior deltoid, latissimus dorsi, teres major, pectoralis major, and triceps brachii act as antagonists for shoulder flexion, the iliopsoas for hip extension, and the hamstring group for knee extension. To maintain this position, it is essential to activate the deep abdominal muscles, such as the TrA muscle [35]. McGill and Karpowicz [40] state that performing the bird dog exercise with hand or foot movements can activate multiple muscle groups. Studies have shown that asking people to lift either their upper or lower limb is a greater challenge to core muscle control [16,41]. McGill [40] includes this exercise as one of the ‘Big 3’ stabilization exercises that are the most appropriate for spine stability.

Regarding the IO, the side plank exercise showed the most significant difference in relative muscle thickness, while the dead bug exercise showed the least (refer to Results). McGill [40] also includes this exercise as one of the ‘Big 3’ stabilization exercises. According to Youdas et al. [42], a biomechanical analysis of the side plank exercise shows that there are three main forces acting on the body: gravity acting on the center of mass and two ground reaction forces acting in opposite directions at the points where the body contacts the ground. These forces tend to cause the body to sag or bend. The muscles on the weight-bearing side are responsible for contracting against the effects of gravity and body mass to maintain proper alignment. The IO muscle is essential for stabilizing the spinal column by directing the load and responding to upper and lower limb movement [17]. The central nervous system (CNS) predicts the effects of movement and plans the recruitment of muscles to overcome disturbances. This prediction is based on an internal system of body dynamics [43]. Side planks activate the internal and external obliques, stabilizing the spine and pelvis and aiding in rotational movements during daily activities [40].

Likewise, the LM relative thickness was greater during the side plank exercise and the lowest during the plank exercise (refer to Results). Although small, multifidus muscles aid several movements of the vertebral column; when contracting bilaterally, they extend the spine, while unilateral contraction like the side plank exercise aids the lateral flexion of the spine to the same side and the rotation of the spine to the opposite side [4]. Similar to the findings of previous studies [44,45,46], the side plank exercise has been shown to activate the lumbar erector spinae and LM muscles.

This study’s results provide partial support for the second hypothesis. It was found that more than one exercise had a significant difference in overall relative muscle thickness while performed with the hollowing maneuver compared to the other exercises (Figure 4). The total thickness of all deep core muscles did not differ significantly between the bird dog, side plank, dead bug, and beast crawl exercises. However, the bird dog and side plank exercises showed significantly higher overall increases in relative muscle thickness compared to the plank, toe tap, and bridge exercises. Meanwhile, the dead bug and beast crawl exercises did not show any significant difference in overall relative muscle thickness from the other three exercises (plank, toe tap, and bridge). Bird dog and side plank exercises are commonly used in core stability programs [19], as well as Pilates [47]. These exercises have been shown to effectively increase the thickness of deep core muscles through activation maneuvers [48]. In our study, two ultrasound machines were used together. The thickness of the TrA, IO, and LM changed simultaneously and had the greatest overall increase during the bird dog and side plank exercises with the hollowing maneuver. Hoseinpoor et al. [49] found that ultrasound measurement established muscle activation relationships between each of the lateral abdominal wall muscles (TrA, IO, and EO) and the LM. Several studies have shown a relationship between TrA, IO, and LM muscles, either as a muscle synergy (% change in thickness) [50] or as a muscle pattern [51,52]. Based on our findings, the following exercises using the hollowing maneuver are ranked in order of their effectiveness in increasing core muscle (TrA, IO, and LM) thickness: bird dog, side plank, beast crawl, dead bug, bridge, toe tap, and plank.

Over the past few years, there has been a debate about the most effective motor control strategy to achieve the dynamic stabilization of the core [40,53,54,55,56]. This study’s findings confirm the third hypothesis as the average relative thickness of all three core muscles (TrA, IO, LM) was greater when using the hollowing compared to the bracing maneuver (Figure 5), which is consistent with previous studies [15,32]. Some studies have suggested that deep abdominal muscles are activated to a greater extent by abdominal hollowing, while superficial muscles are activated to a greater extent by abdominal bracing [26,32]. Other studies, however, have reported exactly the opposite [14,27,57]. Variations in research findings may be attributed to differences in methodology and sample demographics. For example, several studies have examined individuals with LBP or untrained individuals [14,17,19,58]. As hollowing may be more difficult to understand than bracing [32,59,60], this alone could bias the results in favor of one technique.

This study collected and compared exercises from previous studies that demonstrated the greatest thickness of deep core muscles. Additionally, two exercises recommended by exercise specialists, beast crawl and toe tap, were included.

This study’s results have implications for individuals engaging in recreational exercise. These results contribute to the ongoing debate regarding the most effective motor control strategy for the dynamic stabilization of the core [14,15,28,29]. Specifically, for those looking to enhance core stability, the bird dog and side plank exercises, when combined with the hollowing maneuver, may be particularly advantageous. Additionally, this study may have implications for rehabilitation programs, particularly for patients recuperating from injuries or surgeries related to the core muscles. The exercises that demonstrated the most significant increase in muscle thickness could potentially assist in quicker and more efficient recovery. Coaches and trainers may find it beneficial to incorporate the exercises outlined in our findings into their training routines, as core stability is often crucial in many sports and can improve athletes’ performance. Our research could provide evidence-based guidance for developing new fitness classes or regimes focused on core stability, potentially influencing the health and fitness industry. It is important to note, however, that individual differences, proper exercise execution, and other factors can influence the effectiveness of these exercises.

There are limitations to this study that need to be acknowledged. This study conducted a comparison of stability exercises for three core muscles using two activation methods. No intervention was performed in this protocol to observe changes in muscle thickness over time. The results are applicable only for trained physically active young individuals. The practical application of our results is limited to the three muscles that were examined and cannot be extended to other muscles, such as the quadriceps and other trunk muscles. The change in muscle thickness observed through ultrasound provides an indication of muscle size and contraction during exercise [61]. Therefore, our results cannot be directly compared to similar studies conducted with electromyography or MRI. Finally, due to the measurement requirements, the speed execution of all exercises was slow. With in vivo core stability training, participants perform movements at a faster pace.

## 5. Conclusions

The results indicate that the hollowing maneuver led to a greater increase in the thickness of the TrA, IO, and LM across all seven exercises tested, compared to the bracing maneuver. Additionally, the thickness of each muscle reached its maximum value at a different exercise than the other muscles. Of the seven exercises, the bird dog and side plank combined with hollowing showed the greatest increase in overall muscle thickness suggesting that these exercises could be particularly useful for core stability training.

## Figures and Tables

**Figure 1 jfmk-09-00070-f001:**
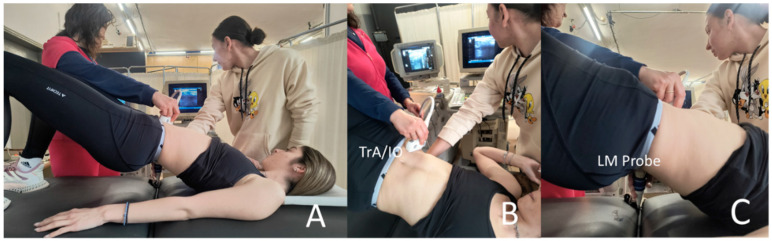
An illustration of the simultaneous measurement of the (**A**) TrA, IO, and LM by means of two ultrasound devices. (**B**) TrA/IO probe. (**C**) LM probe.

**Figure 2 jfmk-09-00070-f002:**
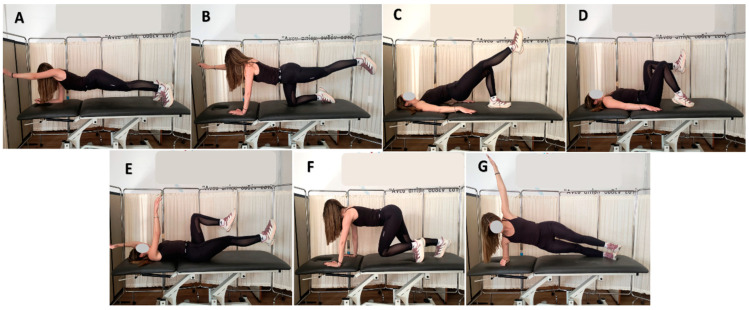
The exercises examined were illustrated with the subject’s consent: (**A**) plank with one arm extended, (**B**) bird dog, (**C**) bridge with one leg extended, (**D**) toe tap, (**E**) dead bug, (**F**) beast crawl, and (**G**) side plank with extended arm.

**Figure 3 jfmk-09-00070-f003:**
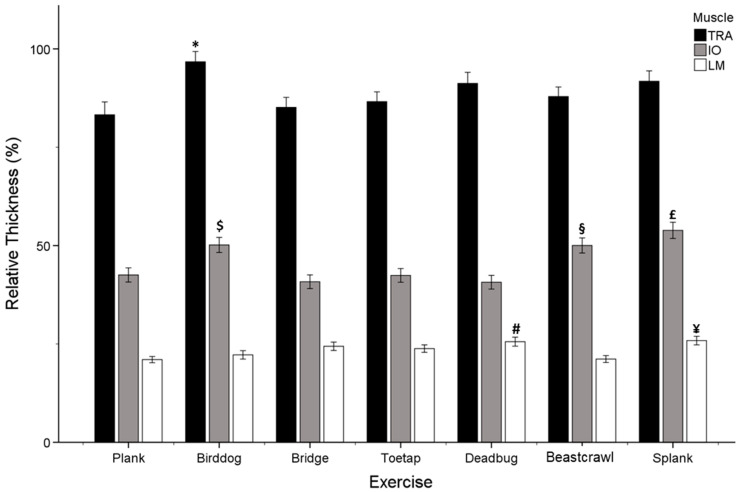
Relative muscle thickness of transversus abdominis (TrA), internal oblique (IO), and lumbar multifidus (LM) for all seven exercises. * = statistically significant in relative thickness from bridge and plank. $, £ = statistically significant in relative thickness from bridge, plank, toe tap, and dead bug. § = statistically significant in relative thickness from toe tap. ¥, # = statistically significant in relative thickness from beast crawl and plank.

**Figure 4 jfmk-09-00070-f004:**
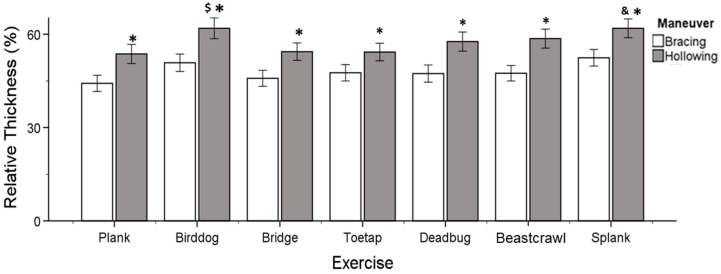
Exercise by maneuver interaction as shown by total mean relative thickness score. * = statistically significant from bracing. $, & = statistically significant in relative thickness from plank, bridge, and toe tap.

**Figure 5 jfmk-09-00070-f005:**
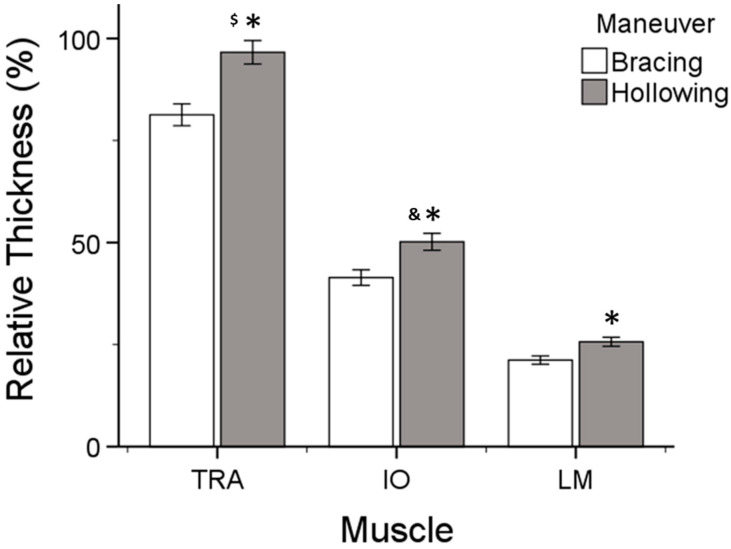
Relative muscle thickness by maneuver of transversus abdominis (TrA), internal oblique (IO), and lumbar multifidus (LM) averaged for all exercises. * = statistically significant from bracing. $ = statistically significant in relative thickness from IO and LM. & = statistically significant in relative thickness from LM.

**Table 1 jfmk-09-00070-t001:** Mean (±SD) age, height, and weight characteristics for total sample and by gender. (BMI = body mass index). N = sample size.

		Age	Height	Weight	BMI
		M ± SD	M ± SD	M ± SD	M ± SD
Total Sample	N = 44	27.0 ± 3.03	1.71 ± 0.08	64.4 ± 11.4	21.9 ± 2.34
Male	N = 18	28.3 ± 3.05	1.78 ± 0.05	74.8 ± 8.51	23.5 ± 2.10
Female	N = 26	26.1 ± 10.0	1.65 ± 0.06	57.2 ± 6.43	20.8 ± 1.82

**Table 2 jfmk-09-00070-t002:** The muscle relative thickness of the transversus abdominis (TrA), internal oblique (IO), and lumbar multifidus (LM) were measured with US in each exercise using either the bracing or hollowing maneuver. The measurements were expressed as a percentage of the thickness at rest and are presented as the mean (±SD).

Exercise	Muscle	Bracing	Hollowing
Plank	TrA	75.4 ± 27.5	90.1 ± 31.8 *
	IO	37.6 ± 15.0 #	47.5 ± 17.4 *#
	LM Total	19.4 ± 6.77 ˆ 44.1 ± 11.2	22.7 ± 7.80 *ˆ 53.7 ± 13.2 *
Bird Dog	TrA	87.1 ± 20.1	106. ± 24.5 *
	IO	45.5 ± 16.4 #	54.8 ± 18.4 *#
	LM Total	19.9 ± 9.69 ˆ 50.8 ± 9.54	24.5 ± 9.93 *ˆ 61.9 ± 12.1 *
Bridge	TrA	79.2 ± 22.9	91.1 ± 23.5 *
	IO	36.2 ± 14.6 #	45.4 ± 16.8 *#
	LM Total	22.1 ± 9.49 ˆ 45.8 ± 11.1	26.7 ± 9.97 *ˆ 54.4 ± 12.4 *
Toe Tap	TrA	81.3 ± 23.4	91.9 ± 22.2 *
	IO	40.2 ± 16.1 #	44.6 ± 16.5 *#
	LM Total	21.4 ± 7.99 ˆ 47.6 ± 10.2	26.3 ± 9.16*ˆ 54.3 ± 11.4 *
Dead Bug	TrA	82.9 ± 26.1	99.6 ± 24.2 *
	IO	36.4 ± 15.9 #	45.0 ± 15.8 *#
	LM Total	22.8 ± 9.68 ˆ 47.4 ± 13.5	28.3 ± 11.1 *ˆ 57.6 ± 11.3 *
Beast Crawl	TrA	78.1 ± 17.9	97.7 ± 22.9 *
	IO	45.6 ± 17.3 #	54.4 ± 17.8 *#
	LM Total	18.7 ± 7.40 ˆ 47.5 ± 8.69	23.6 ± 11.1 *ˆ 58.6 ± 10.6 *
Side Plank	TrA	84.8 ± 23.6	98.7 ± 24.1 *
	IO	48.3 ± 17.9 #	59.4 ± 19.4 *#
	LM Total	24.2 ± 9.85 ˆ 52.4 ± 11.9	27.5 ± 10.5 *ˆ 61.9 ± 13.3 *

* Greater than bracing, *p* < 0.01; ˆ lower than IO and TrA, *p* < 0.01; # lower than TrA, *p* < 0.01.

## Data Availability

Data are contained within the article and Appendix A.

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
