# Peer review of "A Comparison between Core Stability Exercises and Muscle Thickness Using Two Different Activation Maneuvers"

_jfmk, 2024, doi:10.3390/jfmk9020070_

Round 1

Reviewer 1 Report

Comments and Suggestions for Authors

The study presents ultrasound results obtained from different ‘core’ muscles after different so-called core stability exercises. The information contributes to a growing research area concerning core, stability training and is relevant for exercise professionals offering types of exercises for a range of complaints or sport performance. The manuscript is very well written, and is useful addition to the literature, there are a few additions that could be made to the introduction section especially- these are found below.

Introduction

Line 34-36 a few sentences about the studies mentioned is required. Differences/ similarities of methodology and results. Key results relevant to this study. 

It is also relevant to review the literature about acute exercise and validity of ultrasound to measure muscle function during exercise

Line 41 Is the key outcome about ‘thickness’ rather ‘recruitment’?  

Line 37-49 I also think it pertinent to review the literature in regard to frequency and amount of caused ability training and the effect this has on recruitment patterns and core stability performance.

Line 72 what is specifically meant by ‘perspectives among researchers’? Please be more objective.

Please state objective of your study at the end of the introduction section .

Hypothesis should be stated as this is not entirely clear, and he mentioned on page 341

Results

Between pages 194 and 237, there is much text that should be made concise and clearer, outlining the different exercises and different effects on ultrasound results. With that in mind, emphasising the figures representing the results would be beneficial, thereby reducing the need for text. 

The discussion section is well compiled and easy to read. However, in the limitation section between pages 395 and 404 it should also be stated that this was not a training study. A training study is needed to establish the long-term effects on the muscles investigated as well as the different combinations of the exercise themselves, and the different amounts of exercise used. These all could be influential factors in the ultrasound results observed

Author Response

1st Reviewer – first observation

Line 34-36 a few sentences about the studies mentioned is required. Differences/ similarities of methodology and results. Key results relevant to this study. It is also relevant to review the literature about acute exercise and validity of ultrasound to measure muscle function during exercise

Reply

Thank you for your observation. Studies in rows 41-45 investigated the reliability of deep muscle measurements using ultrasound. All studies, including our own (number 13), found ultrasound to be a highly reliable method for these measurements. Thus, we added the following: “The studies above commonly conclude that ultrasound measurement of trunk muscle thickness is a reliable method that can provide useful information about the function of these muscles”.

1st Reviewer – second observation

Line 41 Is the key outcome about ‘thickness’ rather ‘recruitment’?  

Reply

Thank you for your observation. The ultrasound measurements indicate that the correct answer is thickness, not recruitment. Muscle thickness is a numerical index of muscle, according to Kosuge et al. (2023). Electromyography can be used to obtain recruitment.

Kosuge, T., Yamaguchi, T., & Kumagai, H. (2023). Study of the relationship between muscle thickness and the conducting waves of multichannel surface electromyography. Biomedical Signal Processing and Control, 84, 104983.

1st Reviewer – third observation

Line 37-49 I also think it pertinent to review the literature in regard to frequency and amount of caused ability training and the effect this has on recruitment patterns and core stability performance.

Reply

I am in agreement with this observation. However, to review the literature on the frequency and amount of training-induced fitness and its effect on recruitment patterns and core stability performance, we need to make relevant interventions with specific training protocols and establish their effects on core training. Doing so would require prior knowledge of the best exercises to include in the specific protocol. This is what we are currently planning to do as a continuation of the studies we have done so far.

In defense of my answer, I cite the systematic review by Stuber, K. J., Bruno, P., Sajko, S., & Hayden, J. A. (2014). Core stability exercises for low back pain in athletes: a systematic review of the literature. Clinical Journal of Sport Medicine, 24(6), 448-456. The researchers conclude that: «The quantity and quality of literature on the use of core stability exercises for treating LBP in athletes is low. The existing evidence has been conducted on small and heterogeneous study populations using interventions that vary drastically with only mixed results and short-term follow-up. This precludes the formulation of strong conclusions, and additional high quality research is clearly needed».

1st Reviewer – fourth observation

Line 72 what is specifically meant by ‘perspectives among researchers’? Please be more objective.

Reply

Thank you for your observation. It appears that the word 'perspectives' does not accurately convey the intended meaning. The revised sentence is as follows: 'The results of these studies vary, possibly due to the lack of a specific exercise protocol for core stability and differing goals among researchers from various scientific fields.' Lines 80-82. It is reasonable to expect that exercise specialists, physiotherapists, and kinesiologists may have different goals when conducting a study. However, it is important to note that they may also have common goals.

1st Reviewer – fifth observation

Please state objective of your study at the end of the introduction section .

Hypothesis should be stated as this is not entirely clear, and he mentioned on page 341

Reply

Thank you for your observation. The objective of the study was stated on lines 82-84. As for the second hypothesis, we have revised it as follows: 'Additionally, we hypothesize that one of the seven protocol exercises, when combined with the hollowing maneuver, will result in a significant difference in overall relative muscle thickness compared to the other exercises.' Lines 86 – 88.

1st Reviewer – sixth observation

Results

Between pages 194 and 237, there is much text that should be made concise and clearer, outlining the different exercises and different effects on ultrasound results. With that in mind, emphasizing the figures representing the results would be beneficial, thereby reducing the need for text.

Reply

Thank you for your observation. We have removed lines 205-207, 213-215, and 227-229 as they contained non-statistically significant relationships between the exercises. We have also included a statement that the results of statistically significant relationships between the exercises are shown in Figure 3. Line 235

1st Reviewer – seventh observation

The discussion section is well compiled and easy to read. However, in the limitation section between pages 395 and 404 it should also be stated that this was not a training study. A training study is needed to establish the long-term effects on the muscles investigated as well as the different combinations of the exercise themselves, and the different amounts of exercise used. These all could be influential factors in the ultrasound results observed

Reply

Thank you for the observation. The following text has been added to the limitations of the survey: “This study conducted a comparison of stability exercises for three core muscles using two activation methods. No intervention was performed in this protocol to observe changes in muscle thickness over time”. Lines 404 – 407.

Reviewer 2 Report

Comments and Suggestions for Authors

The study meets the research methodological criteria, from summary, introduction, methods, results, discussion and conclusion.

Introduction

It seems fine to me, however it is important to order the logical order of an introduction, therefore I suggest ordering according to these paragraphs

1 paragraph: General considerations of core training

2 paragraph: Applications of core training in quality of life and sports performance

3rd paragraph: In this section you should focus on justifying the study and addressing the problem statement of the study.

4 paragraph: Study objectives and hypotheses

Methodology

It seems to me that it brings together all the sections in a pertinent way. However, it must be improved in

- The absolute or relative reliability of the instrument does not appear, nor does it appear whether this instrument is commonly used to evaluate these exercises (applicability)

- Although figures are added to represent the exercises, the procedures are not clear to me, I suggest being more explicit with the step by step, so that in the future other researchers replicate this same model in other corners of the world, a figure can be added explanatory giving reference to each step of the study.

Statistic analysis

-The absolute or relative reliability of the measurements of each muscle group is not incorporated, how would this be justified?

Results

They seem correct to me

Discussion

It seems correct to me

Conclusion

It seems correct to me

Author Response

2nd Reviewer – first observation

Introduction

It seems fine to me, however it is important to order the logical order of an introduction, therefore I suggest ordering according to these paragraphs

1 paragraph: General considerations of core training

2 paragraph: Applications of core training in quality of life and sports performance

3rd paragraph: In this section you should focus on justifying the study and addressing the problem statement of the study.

4 paragraph: Study objectives and hypotheses

Reply

Thank you for your observations. However, it is important to note that this study focused on core stability training rather than core training. Core stability training requires not only strength but also activation of the central nervous system to coordinate muscles and achieve overall trunk stability during daily and athletic movements.

In 1st paragraph we added the following: “Many scientists acknowledge the importance of core stability training in stabilizing and building strength for both sports and daily activities”.
In 2nd paragraph we added the following: A metanalyses by Dong et al [2] concluded that Core stability training had a large effect on athletes' core strength and balance, but little effect on specific athletic performance. In addition, core stability training was more effective than general exercise at reducing pain and improving back-specific functional status in patients with low back pain [3]. Lines 29 – 34.

3rd paragraph. We justifying the study and addressing the problem statement of the study between lines 59 to 78 and also lines 79 to 82.

4th paragraph. We have study objectives and hypotheses on lines 82 to 90.

2nd Reviewer – second observation

Methodology

It seems to me that it brings together all the sections in a pertinent way. However, it must be improved in

- The absolute or relative reliability of the instrument does not appear, nor does it appear whether this instrument is commonly used to evaluate these exercises (applicability)

- Although figures are added to represent the exercises, the procedures are not clear to me, I suggest being more explicit with the step by step, so that in the future other researchers replicate this same model in other corners of the world, a figure can be added explanatory giving reference to each step of the study.

Reply

Thank you for your observations.

-The absolute or relative reliability of the instrument is well established by many studies, some of which are presented on lines 41 - 45. Ultrasound is a reliable tool for accessing deep core muscle thickness and relative thickness. Refer to our study [13].

- Since the beginning, we have uploaded a supplementary text (Lines 180- 181 S1) explaining how to perform each exercise of the protocol. Please refer to this supplementary section for more detailed information on each exercise.

2nd Reviewer – third observation

Statistic analysis

-The absolute or relative reliability of the measurements of each muscle group is not incorporated, how would this be justified?

Reply

Thank you for your observations. The reliability of the ultrasound measurements was not one of the aims of our study. We have already investigated this in our previous study "Muscle thickness during core stability exercises in children and adults. J. Hum. Kinet. 2020, 71, 131-144, doi:10.2478/hukin-2019-0079" and we found a high reliability of the device measurements (ICC3, 3= 0.76-0.99).

Reviewer 3 Report

Comments and Suggestions for Authors

This study brings practical application knowledge to physiotherapists and trainers. The work is well written, and the methodology is appropriate. However, minor revisions are necessary.

Method

The sample calculation needs to be clarified. Which study (or data) was used as a basis for sample calculation? Which outcome was used to calculate the sample?

Results

The section presenting the results is hard to comprehend and requires improvement. The outcomes should be more specific and clear. For instance, instead of stating that "the beast crawl exercise differed significantly from that in the toe tap exercise" in line 218, it would be better to mention whether "the relative thickness" was significantly higher or lower.

Also, in line 225, it says, "In relation to the LM muscle, the side plank exercise had a statistically significant effect compared to the beast crawl (p < .001) and the plank (p = .027), according to the Bonferroni post- hoc analysis.". What variable are the authors referring to? What effect was identified? It is necessary to correct this issue (besides others) in the result section

Fig 3, 4, and 5. What do the asterisks mean?

Line 52- TrA needs to be written in full.

Line 61- Moghadam et al. there is no number for references.

Line 336- "In similar way as found in this study, many studies" is out of context.

Author Response

3rd Reviewer – first observation

Results

The section presenting the results is hard to comprehend and requires improvement. The outcomes should be more specific and clear. For instance, instead of stating that "the beast crawl exercise differed significantly from that in the toe tap exercise" in line 218, it would be better to mention whether "the relative thickness" was significantly higher or lower. Also, in line 225, it says, "In relation to the LM muscle, the side plank exercise had a statistically significant effect compared to the beast crawl (p < .001) and the plank (p = .027), according to the Bonferroni post- hoc analysis.". What variable are the authors referring to? What effect was identified? It is necessary to correct this issue (besides others) in the result section

Reply

Thank you for your observations. We have added the phrase “greater relative thickness” in lines 219, 229, 231 and 265, the phrase “relative IO thickness” in line 221, and the phrase “relative thickness” in line 267.

3rd Reviewer – second observation

Fig 3, 4, and 5. What do the asterisks mean?

Reply

Thank you for your observation. You are totally right. We forgot to mention that the asterisks declare the statistical significance between the variables. Additionally, we presented the figures using symbols instead of lines and provided corresponding explanations. Lines 245 -260, 273 – 282 and 290 – 302.

3rd Reviewer – third observation

Line 52- TrA needs to be written in full.

Line 61- Moghadam et al. there is no number for references.

Line 336- "In similar way as found in this study, many studies" is out of context.

Reply

Thank you for your observation. You are totally right. We have corrected all the above – Line 61, line 71, and line 346.
